# Achieving green agricultural development: Analyzing the impact of agricultural non-point source pollution on food security and the regulation effect of environmental regulation

Ming Xu[1,2], Zhaoyang Lu[1]*

**1** School of Economics, Southwest University of Political Science and Law, Chongqing, China, **2** School of National Security, Southwest University of Political Science and Law, Chongqing, China

* luzhaoyang@swupl.edu.cn

## Abstract

Food security is the lifeline of national security. It is not only an important cornerstone for world peace, stability, and development, but also the core driving force for promoting the green development of agriculture. This paper utilizes the empirical data of 30 provinces in China from 2010 to 2022 to explore how agricultural non-point source pollution (ANSP) affects food security (FS), and also discusses the moderating effect of environmental regulations (ER). Empirical research shows that the intensification of agricultural non-point source pollution will lead to a decline in the level of food security, especially in the western areas, grain-producing areas, and grain-balanced areas. Furthermore, environmental regulations have a positive moderating effect on the impact of agricultural non-point source pollution on food security. From the perspective of the spatial spillover effect, the inhibitory effect of agricultural non-point source pollution on food security has a spillover effect in space and will radiate to the development of food security in adjacent areas. The research suggests that the local government should enhance its attention and supervision over agricultural non-point source pollution, integrate agricultural technology with environmental supervision, optimize the allocation of agricultural resources, and attach importance to the protection of the agricultural ecological environment, so as to better ensure food security and achieve green agricultural development.

## 1. Introduction

Food security is not only an important part of people's lives, but also an important cornerstone for the healthy, stable, and sustainable development of the economy and society. As a leading agricultural nation, China exhibits distinct traits of heightened carbon emissions and pollution during its agricultural modernization journey. Statistics from the 2020 "Second National Pollution Source Census Bulletin" reveal

**Data availability statement:** Data derived from public domain resources. The data presented in this study are available in China Statistical Yearbook [https://data.stats.gov.cn/easyquery.htm?cn=E0103]. Meanwhile, I uploaded the data set.

**Funding:** This research was supported by the General Project of the National Social Science Foundation of China (Grant No. 23BGL203, awarded to Prof. Zhaoyang Lu). The funders had no role in the study design, data collection and analysis, decision to publish, or preparation of the manuscript.

**Competing interests:** The authors have declared that no competing interests exist.

that agricultural water pollution sources in China discharged 10.67 million tons of chemical oxygen demand, 1.41 million tons of total nitrogen, and 0.21 million tons of total phosphorus, contributing 49.77%, 46.52%, and 67.21% respectively to the country's overall water pollutant discharge. Due to the fundamental position of agriculture in China and the particularity of the uses of agricultural products, as of 2021, the net application of nitrogen, phosphorus and potassium compound fertilizers per mu for the three major food crops in China was 8.39 kilograms, 0.44 kilograms, 0.56 kilograms and 15.54 kilograms respectively, and the utilization rates of chemical fertilizers and pesticides reached 41.3% and 41.8% respectively. It has caused high-intensity and overloading pollution to soil and water quality, further confirming that agricultural non-point source pollution is the main source of agricultural pollutants in China [1]. Currently, with increasing attention to FS, it plays a significant role in ensuring the supply of agricultural products, providing food and nutrition to prevent hunger, and stabilizing social development. By the end of 2023, China's grain output had remained stable at over 1.3 trillion kilograms for nine consecutive years, with a per capita grain availability of 493 kilograms, far exceeding the internationally recognized FS threshold [2]. However, the swift progress of the grain industry is inherently hampered by the repercussions of the agricultural environment. In the context of China's extensive agricultural economic framework, the problem of ANSP, stemming from market entities' relentless pursuit of high-input, high-output, intensive, and large-scale agricultural development, continues to persist as a pressing concern.

In the backdrop of prevailing resource scarcity and profound environmental pollution, safeguarding FS has emerged as a paramount priority. Despite China's policy framework and substantial financial backing aimed at mitigating pollution, the administratively steered financial measures have encountered a "malfunctioning" impasse, particularly in light of the unique characteristics of ANSP. This makes controlling ANSP a critical and challenging aspect of environmental governance [3]. On the one hand, the dispersal, concealment, and lagging effects of ANSP exacerbate the situation, resulting in increasing government financial support year by year, thus necessitating the expansion and supplementation of tools to manage ANSP. On the other hand, amidst the rapid development of the grain industry, there is a growing emphasis on the rational allocation of agricultural resources. This necessitates a shift towards green and low-carbon projects, fostering the transformation of the traditional grain industry into an ecologically sustainable one. Furthermore, the development of new green grain industries is crucial to better safeguard FS. In this context, ANSP, as an important driver of systemic environmental pollution, plays a crucial role in achieving sustainable development of the food industry and ensuring human health and safety [4].

Although ANSP has a certain inhibiting effect on FS, it is also a problem that must be solved at present, including the unbalanced and inadequate development of different regions and production areas in China. Therefore, in this context, it is necessary to study the effect of ANSP on FS. Does it have a substantial inhibitory effect on FS? What mechanism does ANSP use to regulate its effect on FS? The purpose of this paper is to explore scientific research methods and evaluate the potential

mechanisms of the effects of ANSP on FS. This endeavor aims to furnish a comprehensive decision-making framework for governments engaged in ecological development, while also offering invaluable insights for pollution control initiatives and ensuring FS in developing nations. This study aims to promote the improvement of the FS governance system and realize the dual benefits of national security and the environment.

To sum up, the structure of this study is as follows: The first part is the introduction, which presents the research background of this paper; The second part is a literature review, introducing the existing research; The third part is theoretical analysis and research hypotheses, introducing the research mechanism of this paper. The fourth part is the research design, which introduces the measurement methods of ANSP and FS. The fifth part is the empirical results and analysis, introducing the relationship between ANSP and FS. The sixth part is the conclusion and policy implications, providing corresponding policy suggestions for ANSP and FS based on empirical results.

## 2. Literature review

### 2.1. Development of agricultural non-point source pollution and food security

Pollution from agricultural surface sources refers to ecological pollution caused by excessive chemical inputs to the plantation industry, as well as the plantation industry's mishandling of crop residues and livestock manure. Factors such as rainfall, topography, and myriad influences pose significant challenges in monitoring. Amidst the rapid progression of agriculture in the 21st century, China's major lakes and rivers have fallen prey to escalating surface pollution, culminating in the perilous issue of eutrophication [5]. Similarly, foreign countries are facing pollution from agricultural surface sources due to the widespread use of chemical fertilizers and the intensification of modern agricultural practices [6].

The notion of food security has progressed alongside economic and societal advancements. In 1974, the Food and Agriculture Organization of the United Nations (FAO) articulated the objective of food security as "the capacity for every individual, at all times, to access sufficient food for their sustenance and wellbeing," emphasizing that access to adequate food solely for subsistence was the paramount consideration for food security [7]. In 1983, the price factor was introduced, and food security was not only about feeding people, but also about making food affordable [8]. In 1996, the goal of food security was further expanded to include food hygiene, health standards, and nutritional balance [9]. It can be seen that the perception of food security has undergone a shift from "availability" to "goodness".

### 2.2. Methods for measuring agricultural non-point source pollution and food security

It was found that the academic approach to measuring the sources of ANSP is usually based on experimental methods. Sun et al [10] used the literature analysis method as well as an assessment of the provincial nitrogen residual balance on the growth of ANSP. Xu et al [11] used NH3N as an indicator to characterize ANSP and argued that ANSP has a long-term impact on water quality pollution. Zou et al [12], Geng et al [13], and Zhang et al [14] used chemical oxygen demand (COD), total nitrogen (TN), and total phosphorus (TP) as the three primary indicators of pollution, the study assessed ANSP and revealed a strong correlation with the inherent conditions of agricultural endowment.

Although there is no internationally recognized FS measurement system, some scholars measure FS by food production security, consumption security, and distribution security [15]. Some scholars also measure FS by the Food Supply, Food Access, Food production stability, and Food production sustainability [16]. Cui & Zhong [17] measure FS by the area under cultivation of major crops such as rice, wheat, corn, Soybean, and other major crops to measure FS. Han et al [18] evaluated FS through the intricate interplay between food safety, nutritional wellbeing, and resource security, with a focus on the integrity of China's Nutrition-Resource-Food (NRF) system. Jiang et al [19] measured FS by Quantity security, Economic security, Environmental security, and Circulation security. Taken together, most of the assessment methods of FS in the academic world adopt the indicator measurement approach.

   

## 2.3. Impact of agricultural non-point source pollution on food security

The research finds that the academic literature has proved that agricultural pollution has a substantive impact on food security, mainly focusing on the fact that the generation of agricultural pollution, such as nitrogen, has a strong negative effect on food production, which leads to FS problems [20], and confirms that the excessive use of fertilizers in the process of food production and management not only leads to agroecological and environmental pollution but also threatens FS [21]. In addition, FS is also threatened by the underutilization of agricultural resources, agricultural non-point source pollution, land degradation, and heavy metals [22,23]. Therefore, for the generation of ANSP, on the one hand, it is necessary to control the efficiency of nitrogen utilization to protect the agro-ecological environment while guaranteeing FS [24], and on the other hand, it is necessary to appropriately reduce the use of pesticides, agricultural films, and other pollutants to better protect FS [25]. In summary, despite the well-established fact that agricultural pollutants significantly impact FS, there remains a scarcity of literature directly validating the consequences of ANSP on this crucial aspect.

## 2.4. Research gap

Overall, the research conducted on ANSP and FS appears to be both methodical and exhaustive, yielding significant findings across all pertinent dimensions. However, as the research progresses, certain shortcomings of the existing studies have surfaced. In light of this, the present study utilizes panel data spanning from 2010 to 2022, encompassing 30 provinces in mainland China. The FS index was constructed employing the entropy value method, while the emissions of three primary categories of ANSP—total nitrogen, total phosphorus, and chemical oxygen demand—were assessed using the inventory method. In addition, through the establishment of the regulatory effect model, we empirically analyze the regulatory role of ER in the influence of ANSP on FS, and further analyze its spillover effect at the spatial level. The research contributions of this paper include the following three main aspects:

Firstly, an innovative identification of the causal link between ANSP and FS has been established. Prior research predominantly concentrated on the interplay between individual ANSPs, FS, and other pertinent variables. Meanwhile, this paper delves into the moderating function of governmental environmental regulations within this intricate relationship. It not only uncovers the intricate interactions among these various elements but also contributes to enhancing the management of ANSP. Furthermore, it provides empirical evidence that can aid in optimizing the development of China's FS system.

Secondly, this study expanded the measurement range of ANSP based on the list method and FS based on the entropy value method. Most of the existing studies using the list method only select limited indicators, such as the dosage of chemical fertilizers and pesticides, for calculation, and the selection of indicators is not comprehensive enough. Most of the existing studies on FS have only selected factors such as the sown area of grain and grain yield for calculation. In this paper, three total indicators were selected to calculate ANSP, and four total indicators, namely food production security, supply security, access security, and sustainable security, were selected to calculate FS. Compared with traditional studies, the calculation results are more accurate and can evaluate the relationship between ANSP and FS more comprehensively.

Finally, the regional heterogeneity and spatial spillover effects of ANSP and FS are revealed through comparisons in different geographic regions, different production areas, and at the spatial level. At the theoretical level, it fills the research gap of ANSP in FS governance among different development regions and neighboring geographies. Overall, compared with previous studies, this study provides a more scientific and comprehensive understanding to improve the pollution control and emission reduction effects of ANSP. At the same time, it provides empirical references for promoting and securing FS more effectively.

## 3. Theoretical analysis and research hypotheses

### 3.1. Direct impact of ANSP on FS

Agricultural non-point source pollution is a critical obstacle to the greening of global agriculture, exerts a profound influence on safeguarding and advancing global FS. This impact can be clearly demonstrated in various ways. Firstly, the

creation of ANSP undermines the optimal structure of agricultural resource allocation, thereby diminishing the sustainability of food production. Consequently, this results in the restriction of the development of FS at its source [26]. For example, ANSP, as the main source of pollution in China's grain industry, will lead to an increase in the production costs of high-pollution and high-energy-consuming enterprises during the grain production and operation process. Therefore, to better ensure food security, efforts should be made to curb the generation of pollution. Second, ANSP has a detrimental effect on land, which is a pivotal component of food production and management endeavors. This pollution undermines the quality and efficiency of land utilization, thereby imposing stricter standards for safeguarding FS [27]. Finally, ANSP has put forward more demanding requirements on the technological level of food enterprises, and their technological innovation research and development programs are no longer solely aimed at promoting food production, but rather at utilizing more scientific and environmentally friendly agricultural technologies to promote the transition of the food industry to a large-scale, modern and green industry. This shift not only puts higher demands on grain enterprises but also puts higher demands on future green technology research and development [28]. Therefore, the intricacies and uniqueness of ANSP underscore the protracted and challenging journey towards ensuring FS in China and globally, thereby presenting formidable obstacles to the sustainable development of the grain industry. In light of these considerations, the subsequent hypotheses are posited in this paper:

Hypothesis 1 (H1): ANSP will suppress FS.

### 3.2. The moderating role of environmental regulation in curbing FS through ANSP

Existing studies have demonstrated that reinforcing government environmental regulations may initially lead to a decrease in resource utilization efficiency [29]. However, acknowledging the crucial importance of such regulation is imperative in enhancing the efficacy of environmental pollution control efforts [30]. This is evident in two pivotal aspects: First, ecology establishes a cost-driven comparative advantage across various agricultural sectors, thereby facilitating the effective management and mitigation of non-point source pollution. Specifically, enterprises engaged in highly polluting agricultural production are subject to increased "environmental taxes," subsequently resulting in elevated production costs. These sectors frequently lack the research and development capabilities required to swiftly embark on technological innovations, thereby diminishing their competitive edge. On the contrary, green agriculture, endowed with an intrinsic competitive edge rooted in its eco-friendly nature, can alleviate environmental costs by streamlining resource allocation and fostering technological advancements [31]. Secondly, technological advancements are fostered to expedite the eco-friendly transformation of the food industry. As a result, food companies enhance the quality of factor inputs while eliminating outdated production capacity and fostering emerging technological leaders. This progression fosters the dissemination of novel knowledge, industries, and technologies, thereby promoting the environmentally sustainable development of the regional food industry [32]. Thirdly, the regulation of environmental supervision and control has fundamentally ensured the green development of the entire food industry chain, thereby guaranteeing food quality. Specifically, stringent environmental regulations mitigate ANSP, foster the enhancement of food's green productivity, contribute significantly to the safeguarding of food quality and safety, and propel the green evolution of the food industry while robustly ensuring that national FS remains unscathed by environmental hazards [33]. Based on these considerations, the following hypotheses are proposed in this paper.

Hypothesis 2 (H2): ER has a moderating effect on the development of ANSP inhibition of FS.

### 3.3. Regional heterogeneity of ANSP and FS effects

The disparities in economic development, resource allocation, and industrial makeup among various regions in China are also reflected in significant differences in their grain production areas. First, ANSP may further hamper the development of FS in economically disadvantaged areas. On the one hand, in areas with poor economic conditions, the inhibiting effect

of ANSP on FS may be more obvious due to the lower starting point of environmental governance. Therefore, ANSP is more severe under worse environmental conditions, resulting in FS not receiving better protection [34]. On the other hand, backward regions do not have advanced environmental technologies and practices to mitigate ANSP. In this situation, the backward regions do not have the necessary capital and support to integrate advanced technologies, which makes ANSP a significant resistance factor in the development of FS. Second, ANSP potentially impedes the advancement of FS in key food-producing areas. In the primary grain-producing areas, where grain cultivation serves as the paramount factor, ecological and environmental factors may be ignored. Therefore, in the process of food production, due to the lack of attention to ANSP, ANSP will continue to intensify, and eventually, FS will be threatened, and FS cannot be guaranteed. Based on these considerations, this paper proposes the following hypotheses:

Hypothesis 3 (H3): There is regional heterogeneity in the effect of ANSP on FS.

### 3.4. Spatial spillover effects in the impact of ANSP and FS

Since ANSP and FS have different characteristics in different spaces and geographic areas, under the conditions of the era of high-quality development of agriculture, the intensification of ANSP will inevitably radiate into neighboring areas, resulting in nearby areas being affected by ANSP, which in turn affects FS [35]. First of all, in the short term, as pollution continues to intensify, it will lead to the food industry being subjected to environmental distress, which is not conducive to the utilization of the local food industry's resource allocation advantages; in the long term, it is difficult to drive the technological change in the backward areas, which is not conducive to the development of green technology. In addition, ANSP is subject to the endowment conditions of each region, which may further exacerbate the imbalance in the development of the regional food industry, leading to the emergence of a "digital divide" in the development of the food industry, making it difficult to guarantee FS in different regions. Moreover, the level of environmental and food infrastructure construction among regions will directly affect the development of ANSP and FS, especially the huge difference between the infrastructure level of developed and underdeveloped regions, which also leads to a significant difference in the food industry in different regions. Based on this, this paper proposes the following hypotheses:

Hypothesis 4 (H4): There is a spatial spillover effect of ANSP on FS.

## 4. Data and methodology

### 4.1. Selection of variables

**4.1.1. Explained variables.** Food security (FS), and to construct an indicator system of China's FS level based on published literature [36,37]. In this paper, we intend to select corresponding indicators from four dimensions, namely, food supply security, food production security, food access security, and sustainable food security, and measure them using the entropy method, and the specific indicator system is shown in Table 1.

**4.1.2. Core explanatory variables.** Agricultural Non-Point Source Pollution (ANSP). This paper, drawing upon published literature [38,39], employs the unit survey method to quantify ANSP in various regions. The specific pollutants considered in this analysis include total nitrogen (TN), total phosphorus (TP), and chemical oxygen demand (COD). Furthermore, based on the rationality and scientificity of the data on agricultural non-point source pollution, K (potassium) and BOD (Biological Oxygen Demand) are excluded in this paper. The main reasons are: The core of agricultural non-point source pollution lies in controlling agricultural eutrophication, toxic substances, and physical damage. However, due to problems such as weak migration and poor detection adaptability of K and BOD. Therefore, in this paper, the main investigation units of agricultural non-point source pollution are concentrated in three categories: farmland fertilizer pollution, livestock and poultry breeding pollution, and farmland solid waste pollution. The specific calculation methods are as follows:

                                                                                

**Table 1. System of indicators of the level of FS.**

| Target level | Primary indicators | Secondary indicators | Tertiary indicators (units) | Weights | Quality |
|---|---|---|---|---|---|
| FS | Food supply security | Volatility of total food production | (Total food production in the current year – average of total food production in the last five years)/total food production in the current year (%) | 0.0119 | - |
| | | Land mobility | Cultivated land per capita (mu/person) | 0.0474 | + |
| | | Grain reserves | Total grain production (tons) | 0.0538 | + |
| | | Resilience of crops to disasters | Area affected by crops/area sown with crops (%) | 0.0039 | – |
| | | Food supply stability | Grain purchases (tons) | 0.1015 | + |
| | | Food Circulation | Grain sales (tons) | 0.0636 | + |
| | Food production security | Stability of food production | Grain sown area (millions of hectares) | 0.0502 | + |
| | | Level of financial support for agriculture | Grain sown area/Total sown area (%) | 0.0157 | + |
| | | Agricultural innovativeness | Total power of cropland machinery (10,000 kilowatts) | 0.0500 | + |
| | | Level of human capital | Qualified food workers (persons) | 0.0790 | + |
| | | Agricultural productivity | Gross agricultural output/primary sector employment (yuan/person) | 0.2955 | + |
| | | Infrastructure development | Number of agricultural water conservancy facilities constructed (units) | 0.0755 | + |
| | Food access security | Rural Engel coefficient | Rural food consumption expenditure/total consumption expenditure (%) | 0.0048 | – |
| | | Food price volatility | (Current year food price index – previous year food price index)/ Current year food price index (%) | 0.0043 | – |
| | | Food share | Total food production/resident population (tons/person) | 0.0444 | + |
| | | Road density | Length of transport routes (rail, road, waterway)/urban area (km/sq km) | 0.0531 | + |
| | Sustainable food security | Pesticide application rate | Pesticide application per unit of food sown area/crop sown area (%) | 0.0035 | – |
| | | Fertilizer application rate | Fertilizer application per unit of food sown area Crop sown area (%) | 0.0081 | – |
| | | Agricultural film use | Agricultural film uses per unit of food sown area/crop sown area (%) | 0.0040 | – |
| | | Quality assurance | Effective irrigated area/crop sown area (%) | 0.0301 | + |

$$Pollution_c^t = \sum_\gamma Po_{c\gamma t} \times factor_{c\gamma 1} \times factor_{c\gamma 2}$$

(1)

Equation (1): $Pollution_c^t$ represents the agricultural non-point source pollutant emissions of c province in t years, including TN, TP, and COD, respectively; γ denotes the various types of investigation units; $Po_{c\gamma t}$ denotes the number of each investigation unit in region c in year t; $factor_{c\gamma 1}$ denotes the loss coefficients of each investigation unit; and $factor_{c\gamma 2}$ denotes the coefficients of pollution production of each investigation unit (Table 2).

In this paper, the principal component analysis method is used to downscale the ANSP in the non-expected output variables. At the same time, in order to avoid a negative or zero value situation, after the dimensionality reduction to get the composite factor score, the data after the principal component dimensionality reduction will be converted to the values in the interval of [0, 100], and the conversion method is shown as follows:

$$ANSP_c = [S_c/(\max S_c - \min S_c) \times 0.4 + 0.6] \times 100$$

(2)

**Table 2. ANSP investigation unit.**

| Pollution source | Investigation unit | Survey index | Unit | Discharge inventory |
|---|---|---|---|---|
| Field fertilizer | Nitrogen fertilizer, phosphate fertilizer | Apply refraction | Ten thousand tons | TN、TP |
| Livestock and poultry breeding | Cattle, sheep, pigs, poultry | Stock/stock output | Ten thousand, ten thousand heads | TN、TP、COD |
| Field solid waste | Potato, soybean, corn, rice, wheat | yield | Ten thousand tons | TN、TP、COD |

Equation (2): $ANSP_c$ denotes the agricultural non-point source pollution index of province c, $S_c$ denotes the composite factor score of province c, $\max S_c$, $\min S_c$ and denotes the maximum and minimum values of the composite factor score, respectively.

**4.1.3. Control variables.** Given the myriad factors influencing the degree of FS, this paper draws upon existing published literature for insights [40,41]. The following control variable is set: The urbanization rate (URBAN) is expressed as the proportion of urban population to the permanent resident population in each province. Government intervention (FE) is measured by the proportion of government financial expenditures of each province in relation to the Gross Domestic Product (GDP) of the respective province. Foreign Direct Investment (FDI), represented as the logarithmic form of the proportion of actual utilization of FDI in each province's GDP; Industrial Structure (IS), defined by the share of value added from the primary industry in each province's GDP. Human capital (HR), the number of years of education per capita was chosen to measure human capital, with the formula: average years of education = (illiteracy × 1 + number of elementary school education × 6 + number of middle school education × 9 + number of high school and junior college education × 12 + number of college and bachelor's degree or above × 16)/total number of people over 6 years of age.

**4.1.4. Moderating variables.** Environmental Regulation (ER). As China places greater significance on environmental conservation, environmental regulation emerges as a pivotal tool in mitigating ANSP and safeguarding FS. The interplay between these regulatory measures and ANSP, in turn, exerts a notable influence on ensuring FS. Therefore, it is advisable to refer to the published literature [42]. In this paper, the interaction term of ER and ANSP will be selected as the moderating variable of this paper, environmental regulation will be evaluated based on the volume of accomplished investment in industrial pollution control, as well as the proportion of gross regional product and total energy consumption, respectively.

## 4.2. Model setting and variable selection

Based on the aforementioned analysis, a two-way fixed effect model has been devised to validate the influence of agricultural non-point source pollution on food security, utilizing the following formula:

$$FS_{i,t} = \alpha + \beta_1 ANSP_{i,t} + \beta_2 X_{i,t} + \mu_i + \lambda_t + \varepsilon_{i,t} \tag{3}$$

In model (3), FS serves as the explanatory variable, representing food security, while ANSP assumes the role of the core explanatory variable, representing agricultural non-point source pollution. The set of control variables is denoted by X, where i and t represent the specific province and the period, respectively. Additionally, λi denotes the area fixed effect, λt represents the year fixed effect, and εit stands for the random disturbance term.

Furthermore, to delve into the moderating influence of environmental regulation on food security, we incorporated the interaction variable between agricultural non-point source pollution and environmental regulation into the Baseline model (3), resulting in the formulation of Model (4):

$$FS_{i,t} = \alpha + \beta_1 ANSP_{i,t} + \beta_2 (ANSP_{i,t} \times ER_{i,t}) + \beta_3 X_{i,t} + \mu_i + \lambda_t + \varepsilon_{i,t} \tag{4}$$

In model (4), ER signifies the degree of environmental regulation, whereas X embodies a comprehensive array of control variables.

To delve deeper into the implications of agricultural non-point source pollution on the harmonious development of food security, model (3) is extended to a Spatial autoregressive model (SAR), offering a more nuanced perspective:

$$FS_{i,t} = \alpha + \rho WFS_{i,t} + \beta_1 ANSP_{i,t} + \beta_2 X_{i,t} + \mu_i + \lambda_t + \varepsilon_{i,t} \tag{5}$$

In model (5), ρ is the spatial autoregressive coefficient and W is the spatial weight matrix; other symbols have the same meaning as in model (3).

### 4.3. Data sources

This paper is grounded in the reality of ANSP and China's FS status, with the aim of showcasing the accessibility and practicality of the data utilized. The raw data is sourced from reputable publications such as the China Environmental Statistics Yearbook, China Grain and Material Reserve Yearbook, China Statistical Yearbook, China Rural Statistical Yearbook, China Population and Employment Statistical Yearbook, China Agricultural Product Price Survey Yearbook, along with the developmental work reports compiled by various provinces. Considering that the data of each indicator may be missing in different years, this paper sets the time interval of the sample data as 2010–2022 in order to obtain more complete data resources as much as possible and to reflect the latest situation of ANSP and China's FS level, while the sample regions are selected due to the data limitation of Xizang, Hong Kong, Macao, and Taiwan. The remaining 30 provinces (autonomous regions) were selected due to data limitations. However, there are still individual provinces with missing data, and this paper will use linear interpolation to fill in that part. The sign of each variable and the results of the expressive statistical analysis are shown in Table 3:

## 5. Result

### 5.1. Empirical strategy and results

The analysis of variance inflation factors (VIF) in the regression equation revealed that the VIF values ranged from 1.62 to 6.86, which were notably below the critical threshold of 10. This observation underscores the absence of any notable multicollinearity issues among the independent variables. Following the diagnostic tests, this study opted for a two-way fixed effects model for the empirical regression analysis. Specifically, column (1) represents the direct regression between ANSP and FS, while columns (2) through (6) detail the regression outcomes after sequentially incorporating control variables. The comprehensive results are presented in Table 4:

**Table 3. Descriptive statistics of variables.**

| Variable | Symbol | N | Average | SD | Min | Max |
|---|---|---|---|---|---|---|
| Food security | FS | 390 | 0.155 | 0.077 | 0.039 | 0.464 |
| Agricultural non-point source pollution | ANSP | 390 | 0.601 | 0.078 | 0.506 | 0.831 |
| Urbanization rate | URBAN | 390 | 0.594 | 0.123 | 0.338 | 0.895 |
| Financial expenditure | FE | 390 | 0.257 | 0.111 | 0.103 | 0.751 |
| Foreign direct investment | FDI | 390 | 11.935 | 59.558 | 0.769 | 833.705 |
| Industrial structure | IS | 390 | 0.098 | 0.053 | 0.002 | 0.258 |
| Human capital | HR | 390 | 9.341 | 0.890 | 7.606 | 12.697 |
| Level of economic development | GDP | 390 | 26133.670 | 22592.230 | 1144.200 | 127739.400 |
| Industrial investment | IPI | 390 | 199739.500 | 200984.700 | 476 | 1416464 |
| Proportion of energy consumption | EG | 390 | 0.105 | 0.071 | 0.030 | 0.409 |

**Table 4. Baseline regression results.**

| Variable | (1) | (2) | (3) | (4) | (5) | (6) |
|---|---|---|---|---|---|---|
| ANSP | −1.414*** (−3.12) | −0.893** (−2.14) | −0.967** (−2.29) | −1.011** (−2.40) | −1.189*** (−2.79) | −1.120*** (−2.65) |
| URBAN | | −0.987*** (−8.40) | −1.009*** (−8.48) | −0.974*** (−8.16) | −0.941*** (−7.87) | −0.896*** (−7.49) |
| FE | | | −0.076 (−1.19) | −0.059 (−0.92) | −0.067 (−1.05) | −0.049 (−0.77) |
| FDI | | | | −0.026** (−2.20) | −0.026** (−2.19) | −0.027** (−2.34) |
| IS | | | | | 1.092** (2.28) | 1.042** (2.19) |
| HR | | | | | | 0.083*** (2.62) |
| Year | Control | Control | Control | Control | Control | Control |
| Region | Control | Control | Control | Control | Control | Control |
| _Cons | −2.890*** (−12.18) | −3.296*** (−14.86) | −3.466*** (−13.13) | −3.405*** (−12.90) | −3.608*** (−13.02) | −4.241*** (−11.60) |
| $R^2$ | 0.547 | 0.623 | 0.625 | 0.630 | 0.636 | 0.643 |
| N | 390 | 390 | 390 | 390 | 390 | 390 |

Note: t-statistics are in parentheses; *, **, and *** indicate significance at the 10%, 5%, and 1% levels, respectively, as in the following tables.

From the regression analysis presented in Table 4, it is evident that ANSP exhibits a statistically significant negative impact at the 1% level, even without the inclusion of control variables. Upon the addition of these variables, the inhibition of ANSP on FS levels remains significant at the 1% level, thereby confirming Hypothesis H1. Specifically, Column (6) reveals a coefficient of −1.120 for ANSP, significant at the 1% level, indicating a statistically significant hindrance to the growth of FS levels. This translates to a decrease of 1.120 percentage points in FS for every 1 percentage point increase in ANSP. This underscores the profound inhibitory effect of intensified ANSP on FS. Consequently, it is imperative for the government to integrate ANSP into its FS policy framework and prioritize efforts to mitigate this pollution, thereby safeguarding FS through policy guidance. This approach serves as a crucial catalyst for advancing national security strategies and fostering high-quality development within the food industry.

### 5.2. Robustness regression

To further strengthen the credibility of the previous baseline regression findings, the subsequent four approaches are employed for verification.

First, the control variables are lagged by one period, adhering to the research methodology outlined by Miao et al [43]. The robustness test employs a one-period lag method for the control variables, a strategy that efficiently mitigates the endogeneity issue in model estimation. The findings of this test are presented in column (1) of Table 5, revealing that the coefficient of the impact of ANSP on FS remains significantly negative at the 1% level. This underscores the robustness of the benchmark regression results presented in this paper.

Secondly, the core explanatory variables undergo scrutiny via the reduced-tail methodology. To fortify the credibility and resilience of our study, discrete variables are excised from the FS measurement data and reassessed utilizing the baseline regression model [44]. The outcomes of this re-evaluation are presented in column (2) of Table 5. Notably, the coefficient indicating the relationship between ANSP and FS exhibits a statistically significant negative correlation at the 1% level, reinforcing the inhibitory impact of ANSP on the advancement of FS.

**Table 5. Robustness regression.**

| Variable | (1) | (2) | (3) | (4) |
|---|---|---|---|---|
| ANSP | −1.495*** (−3.29) | −1.191*** (−2.78) | −3.804** (−2.37) | −1.065** (−2.51) |
| Control variable | Control | Control | Control | Control |
| Year | Control | Control | Control | Control |
| Region | Control | Control | Control | Control |
| AR (2) | | | 0.532 | |
| Hansen | | | 0.359 | |
| N | 390 | 384 | 360 | 390 |

Thirdly, replace the model. Given the potential for estimation bias and endogeneity issues in the conventional two-way fixed-effects regression model, the System GMM regression model was selected as an alternative model [45]. The test outcomes are presented in column (3) of Table 5, where the coefficient indicating the impact of ANSP on FS remains significantly negative at the 5% level, underscoring the robustness of the baseline regression findings.

Fourth, incorporate a control variable to account for economic development, a pivotal force propelling FS advancements. This not only establishes a solid material foundation for agriculture but also fosters a noteworthy enhancement in overall agricultural efficiency [46]. For this study, we adopted the GDP of each province as an indicator of economic development, subjecting it to logarithmic transformation and subsequently integrating it into our baseline regression model. As depicted in column (4) of Table 5, ANSP continues to exert a substantial negative influence on FS, with its coefficient attaining statistical significance at the 5% level, thereby reinforcing the robustness of our regression findings.

## 5.3. Heterogeneity test

To better understand the effects of ANSP on FS, this paper re-examines and regresses the impacts across various regions and production areas within China, delving into its heterogeneous effects. This not only augments our understanding of the implications of ANSP on FS but also offers a nuanced perspective for future policy crafting.

Due to the substantial disparities in agricultural development, pollution levels, and food cultivation methods across various regions, the influence on FS development may vary significantly. Consequently, we have categorized the sample into three distinct regions: Eastern, Central, and Western, to delve into their unique performance trends. The findings are presented in columns (1)-(3) of Table 6. Notably, the eastern and central regions exhibit an inconspicuous inhibitory effect of ANSP on FS. This can be attributed to their advanced economic and agricultural development, cutting-edge technologies, and stricter agricultural pollution control measures, thereby mitigating the impact of such pollution on FS. Conversely, the western region displays the most pronounced inhibitory effect, highlighting the unique challenges and constraints it faces in food production. The complex terrain, primarily characterized by hills and mountains, coupled with high ecological protection costs and inadequate financial investment and technology, fails to meet the threshold of "high standards." Consequently, ANSP exerts a formidable inhibiting influence on FS in this region.

According to their unique agricultural attributes and geographical contexts, China's provinces have evolved into distinct grain functional zones, encompassing primary grain-producing areas, primary sales regions, and regions where production and sales are in balance [47]. The varied agricultural functional positioning within these zones fundamentally shapes the mechanisms and outcomes associated with FS. Consequently, we categorized the samples into three distinct groups based on the agricultural functional characteristics of each province: grain-producing areas (PA), grain-consuming areas (CA), and grain-balance areas (BA), and performed separate regression analyses. The specific regression outcomes are presented in Table 6, subsections (4) through (6). Within the grain-producing areas, ANSP exerts a pronounced and statistically significant negative impact on FS, at a 5% significance level. This underscores the

**Table 6. Heterogeneity regression.**

| Variable | (1) Eastern | (2) Central | (3) Western | (4) PA | (5) CA | (6) BA |
|---|---|---|---|---|---|---|
| ANSP | −0.825 (−1.08) | −0.863 (−1.30) | −2.594*** (−2.73) | −0.949** (−2.04) | −2.255** (−2.08) | 0.451 (0.46) |
| Control variable | Control | Control | Control | Control | Control | Control |
| Year | Control | Control | Control | Control | Control | Control |
| Region | Control | Control | Control | Control | Control | Control |
| N | 143 | 104 | 143 | 169 | 130 | 91 |

significant inhibitory influence of such pollution on ensuring FS in regions dominated by grain production. Conversely, in grain-balance areas, ANSP has a more pronounced influence on agricultural production efficiency, with a higher inhibition coefficient. This finding suggests that insufficient attention is given to ANSP in grain production and management activities in these areas, contributing to the intensification of its inhibitory effects. In the grain-consuming areas, while the impact on FS exhibits a positive trend, it is not statistically significant. This can be attributed to the relatively limited food production and operational activities in these areas, where the presence of ANSP does not significantly impact FS. Additionally, the existence of ANSP does not have a substantial impact on FS, and at the same time, more attention may be paid to the protection of the food industry due to agricultural pollution, and thus plays a certain role in promoting the grain-consuming areas.

## 5.4. Regulating effect

The preceding theoretical examination demonstrates that the interplay or synergy between ER and ANSP exerts a constructive moderating influence in safeguarding FS. In this study, we initially select the quantum of investment in industrial pollution control as a metric for environmental regulation and assess its interaction with ANSP. The outcomes, as presented in Table 7, column (1), reveal that an escalation in ANSP exerts a notable inhibitory impact on FS. However, the outcomes of the interplay between environmental regulation and ANSP signify a substantial role in bolstering FS. Furthermore, we reassess environmental regulation using the ratio of GDP to total energy consumption and re-examine its interaction with ANSP. The findings, depicted in Table 8, column (2), mirror those observed in column (1). Consequently, it is evident that the interactive effect of ANSP and environmental regulation contributes significantly to FS, which underscores the need for the enhancement of environmental laws and regulations to more effectively realize the vital national strategy of ensuring FS.

**Table 7. Regulating effect regression.**

| Variable | (1) | (2) |
|---|---|---|
| ANSP | −1.089** (−2.59) | −1.331*** (−3.09) |
| ANSP×IPI | 0.017** (2.10) | |
| ANSP×EG | | 1.140** (2.23) |
| Control variable | Control | Control |
| Year | Control | Control |
| Region | Control | Control |
| N | 390 | 390 |

## 5.5. Space spillover effect

To delve deeper into the spatial spillover effect of ANSP on FS, this paper undertakes an additional regression analysis leveraging spatial econometric modeling. However, prior to employing the spatial model, it is imperative to ascertain the spatial correlation between the explanatory and explained variables [48]. This paper validates this through Moran's index test, utilizing the geographic collinearity matrix, with the outcomes presented in Table 8. Across the years 2010–2022, the global Moran index for ANSP exhibits a significant positive correlation at the 10% threshold, while the index for FS level demonstrates a notably positive correlation at both 1% and 5% levels, respectively. Temporally, the spatial correlation between ANSP and FS in China displays fluctuations, which could stem from the unequal distribution of ANSP and FS levels across varying time periods. Given the significant spatial autocorrelation identified between these two variables, it is imperative to factor in this spatial correlation when assessing the impact of ANSP on FS. Consequently, employing a spatial measurement model to quantify the effect of ANSP on FS is a suitable approach.

Secondly, the application of the spatial model in this paper employs the optimal estimation method, specifically the spatiotemporal double fixed-effects SAR model, subsequent to diagnostic tests encompassing Wald, LM, and Hausman tests. The estimation process relies on the geographic collocation matrix, and the outcomes are presented in Table 9. The findings reveal that the coefficient of ANSP 's impact on FS stands at −0.973, displaying a statistically significant negative correlation at the 5% level. This validates hypothesis H4, confirming that ANSP not only impedes the growth of FS within the region but also exerts a notable inhibitory effect across neighboring regions. Moreover, the study demonstrates that

**Table 8. Spatial correlation regression.**

| Year | ANSP | Z-value | FS | Z-value |
|------|------|---------|-----|---------|
| | Moran's I | | Moran's I | |
| 2010 | 0.121* | 1.300 | 0.176** | 1.735 |
| 2011 | 0.120* | 1.295 | 0.177** | 1.751 |
| 2012 | 0.128* | 1.358 | 0.203** | 1.974 |
| 2013 | 0.129* | 1.362 | 0.231** | 2.219 |
| 2014 | 0.127* | 1.347 | 0.253*** | 2.423 |
| 2015 | 0.130* | 1.371 | 0.279*** | 2.728 |
| 2016 | 0.131* | 1.382 | 0.298*** | 2.842 |
| 2017 | 0.123* | 1.314 | 0.278*** | 2.680 |
| 2018 | 0.124* | 1.319 | 0.290*** | 2.829 |
| 2019 | 0.126* | 1.331 | 0.226** | 2.214 |
| 2020 | 0.127* | 1.337 | 0.203** | 2.013 |
| 2021 | 0.129* | 1.351 | 0.252*** | 2.465 |
| 2022 | 0.130* | 1.358 | 0.260*** | 2.507 |

**Table 9. Spatial spillover effect regression.**

| Geographical adjacency weight matrix | Main | Rho | LR_Direct | LR_Indirect | LR_Total |
|---|---|---|---|---|---|
| ANSP | −0.973** <br>(−2.50) | 0.219*** <br>(3.19) | −0.971** <br>(−2.41) | −0.249* <br>(−1.77) | −1.221** <br>(−2.41) |
| Control variable | Control | Control | Control | Control | Control |
| Year | Control | Control | Control | Control | Control |
| Region | Control | Control | Control | Control | Control |
| N | 390 | 390 | 390 | 390 | 390 |

the three facets of ANSP—direct effect, spatial spillover effect, and total effect—all exhibit varying degrees of significant negativity. This underscores the fact that ANSP not only hinders the progress of FS locally but also possesses a spillover effect, adversely affecting the well-being of FS in adjacent regions.

## 6. Conclusion and policy implications

### 6.1. Conclusion

In this study, we have devised a comprehensive system for quantifying ANSP and evaluating FS, systematically tracking their temporal and spatial trends. Furthermore, we delve into the repercussions of ANSP on FS and theoretically unravel the underlying mechanisms that govern this relationship. Leveraging panel data spanning from 2010 to 2022, encompassing 30 provinces in China, we employ a multifaceted approach that integrates two-way panel fixed-effects, moderated-effects, and spatial-effects models to assess the multidimensional impact of ANSP on FS and its intricate mechanisms. The key findings are outlined below:

Benchmark regression reveals that ANSP exerts a notable inhibitory influence on FS, suggesting that the escalating ANSP poses a significant threat to China's FS, thereby partially elucidating why developed nations remain vigilant about this issue and strive to address ANSP. In the context of China, as a major agricultural nation, despite some mitigation in ANSP, serious contamination persists in food production activities. Fundamentally, this underscores the need for greater government attention and comprehensive management capabilities. It serves as a reminder to developing countries that, amidst addressing food scarcity, the challenge of agricultural pollution must not be overlooked.

Heterogeneity analysis shows that in the eastern and central regions, where economic development is more developed, the exacerbation of ANSP has a certain effect on FS, but it is not a major factor. In the western area with less developed economic development, the exacerbation of ANSP has a significant inhibitory effect on FS. In addition, ANSP in both the grain-producing and grain-balancing areas has a significant inhibitory effect on FS, while ANSP in the grain-consuming areas has a certain contribution to FS, but it is not the main factor. On the one hand, ANSP is closely related to economic development, because the control of agricultural pollution and the improvement of infrastructure construction require a large amount of financial investment, so as to better protect the ecological environment and FS. On the other hand, ANSP has different performance in different food-producing regions, because the grain-consuming areas and grain-balancing areas have more food production and less pollution control, while the grain-consuming areas have limited production and pay more attention to environmental protection. For developing countries, it is not advisable to develop food production activities in a particular region, but rather in areas with sufficient financial resources and a higher level of management.

The moderating effect shows that the interaction term between ER and ANSP has a moderating effect on FS, which suggests that the use of ER will promote FS and can better guarantee FS in China. The advantage of environmental regulation is that it can better reduce agricultural pollution and promote the improvement of green production capacity in the process of food production and management, but the challenges of terrain obstacles and policy implementation may affect the effectiveness of ensuring FS. Therefore, in order to fully utilize the advantages of environmental regulation, it is important to implement environmental regulation in a realistic way that is tailored to local resource endowments in order to achieve the desired results.

The spatial spillover effect shows that there is a spatial spillover effect between ANSP and FS, which not only re-validates the robustness of the baseline regression and proves that the exacerbation of ANSP leads to a decrease in FS, but also proves that the effect of the exacerbation of ANSP radiates to other neighboring areas, leading to a decrease in the level of FS in the neighboring areas. This suggests that local governments should make more efforts to control local ANSP, better allocate agricultural resources, reduce waste in the production process, and reduce environmental pollution with the same inputs.

## 6.2. Policy implications

First, bolster policy support and execution oversight for ANSP management. The Government must not only enhance policy incentives for addressing ANSP but also refine implementation guidelines and monitoring frameworks to forge a robust legal framework. Leveraging financial incentives and environmental stewardship initiatives, it should ensure that farmers and agro-enterprises engaged in the food production cycle adopt rigorous supervisory systems, ensuring that pollution emanating from food production adheres to agricultural pollution discharge norms. This approach will effectively safeguard FS while concurrently mitigating ANSP.

Second, promote the deep integration of agricultural technological innovation and land mobility. Technological innovation and land fluidity are pivotal elements in attaining FS. It is advisable for the government to augment its investments in agricultural research endeavors, devise novel and cutting-edge green technologies, and enhance the dissemination and implementation of eco-friendly and efficient technologies, such as precision agriculture and water-saving irrigation systems. Concurrently, refining land transfer policies, optimizing land usage efficiency, safeguarding farmers' land rights and interests, prioritizing land ecological preservation, minimizing pollution emissions, and ensuring the augmentation of the food industry's ecological efficiency are imperative.

Thirdly, optimize the allocation of human resources and deepen the reform of the environmental system. The overall competency of the rural labor force is a pivotal issue that significantly impacts ANSP. The government must refine its policies, wisely steer the labor force towards acquiring knowledge about ANSP, bolster agricultural vocational education and skills training, diminish the expenditure associated with adopting green technology, and elevate farmers' proficiency in applying agricultural green technology. In addition, it should increase the implementation of environmental systems in agriculture, establish better environmental regulatory measures, realize green production in the food industry, and further ensure FS.

## 6.3. Research shortcomings and prospects

Future research can be conducted in two primary dimensions: firstly, enhancing the granularity of the sample data. While the provincial-level sample data employed in this study offers a macro-scale analytical lens, to delve deeper into the specific impacts of ANSP on FS across diverse geographical regions, future endeavors should strive to refine the sample data further, extending its reach to prefecture-level cities or even counties. This approach will facilitate a more precise identification of the varying effects within distinct geographic and economic landscapes. Secondly, conducting a more exhaustive evaluation of policy effects. Despite utilizing a double fixed-effects model to assess the relationship between ANSP and FS, the examination of policy implications remains limited. Hence, future investigations should incorporate a more comprehensive evaluation framework to unravel the long-term consequences and potential repercussions of ANSP on FS.

## Acknowledgments

The corresponding author gratefully acknowledges Professor Jingang Zhao of the University of Saskatchewan for his valuable guidance, and the Department of Economics at the University of Saskatchewan for their hospitality during his visit from January 2024 to September 2025.

## Author contributions

**Conceptualization:** Ming Xu.

**Data curation:** Ming Xu, Zhaoyang Lu.

**Formal analysis:** Ming Xu, Zhaoyang Lu.

**Funding acquisition:** Zhaoyang Lu.

**Investigation:** Ming Xu.

**Methodology:** Ming Xu, Zhaoyang Lu.

**Project administration:** Ming Xu.

**Resources:** Zhaoyang Lu.

**Software:** Ming Xu.

**Supervision:** Ming Xu, Zhaoyang Lu.

**Validation:** Ming Xu.

**Visualization:** Zhaoyang Lu.

**Writing – original draft:** Ming Xu, Zhaoyang Lu.

**Writing – review & editing:** Ming Xu, Zhaoyang Lu.

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
