## [Decision Letter · Decision Letter 0]

15 Apr 2025

PONE-D-24-54342Achieving green agricultural development: Analyzing the impact of agricultural non-point source pollution on food security and the regulation effect of environmental regulationPLOS ONE

Dear Dr. Xu,

Thank you for submitting your manuscript to PLOS ONE. After careful consideration, we feel that it has merit but does not fully meet PLOS ONE’s publication criteria as it currently stands. Therefore, we invite you to submit a revised version of the manuscript that addresses the points raised during the review process.

We look forward to receiving your revised manuscript.

Kind regards,

Murli Dhar Meena, Ph.D

Academic Editor

PLOS ONE

Journal Requirements:

“1. Research on security system of data element allocation in the development of new quality productivity. The funding is supported by Southwest University of Political Science and Law. 2. Research on the security system of cross-border data flow from the perspective of new quality productivity (SHZLQN2404). The funding is supported by the Sichuan Police College. 3. Research on Social security risk early warning and governance mechanism of megacities -- A case study of Chongqing (KJQN202400318). The funding is supported by the Chongqing Municipal Education Commission.”

5. We note that your Data Availability Statement is currently as follows: All relevant data are within the manuscript and in Supporting Information files.

6. We note that Figure 1 and 2 in your submission contain map/satellite images which may be copyrighted. All PLOS content is published under the Creative Commons Attribution License (CC BY 4.0), which means that the manuscript, images, and Supporting Information files will be freely available online, and any third party is permitted to access, download, copy, distribute, and use these materials in any way, even commercially, with proper attribution. For these reasons, we cannot publish previously copyrighted maps or satellite images created using proprietary data, such as Google software (Google Maps, Street View, and Earth). For more information, see our copyright guidelines: http://journals.plos.org/plosone/s/licenses-and-copyright.

 a. You may seek permission from the original copyright holder of Figure 1 and 2  to publish the content specifically under the CC BY 4.0 license. 

Additional Editor Comments:

Thank you for submitting your manuscript to PLOS ONE. After careful consideration, we feel that it has merit but does not fully meet PLOS ONE’s publication criteria as it currently stands. Therefore, we invite you to submit a revised version of the manuscript that addresses the points raised during the review process

Reviewers' comments:

Reviewer's Responses to Questions

**Comments to the Author**

1. Is the manuscript technically sound, and do the data support the conclusions?

Reviewer #1: Yes

2. Has the statistical analysis been performed appropriately and rigorously? 

Reviewer #1: Yes

3. Have the authors made all data underlying the findings in their manuscript fully available?

Reviewer #1: Yes

4. Is the manuscript presented in an intelligible fashion and written in standard English?

Reviewer #1: Yes

5. Review Comments to the Author

Reviewer #1: Xu and Lu reported proposed finding based on empirical data from 30 provinces in China during 2010-2022, and nicely explores how agricultural non-point source pollution affects food security and the regulatory effects of environmental regulations, The study shows that the intensification of agricultural non-point source pollution significantly inhibits the growth of food security levels, especially in western regions, major grain-producing areas, and grain-balanced regions. This study proposes that local governments ought to heighten their focus on and regulation of agricultural non-point source pollution, harmonize agricultural technology with land mobility, refine human resource allocation, and fortify environmental regulatory reforms to safeguard food security. I recommend this paper for publication after minor revisions.

comments

1. Authors should define green agriculture in introduction section as the green agriculture is mentioned in the title.

2. Sentence mentioned in the introduction (first page) In 2021, the net per-acre usage of nitrogen, phosphorus, and potassium compound fertilizers for China's three major grain crops was 8.39 kg, 0.44 kg, 0.56 kg, and 15.54 kg, respectively needs to be check again. it seems some typographical error.

3. A total 11 tables have been presented in the paper, therefore authors should present one or two table in figure format.

4. Authors should justify why Total K (potassium) and BOD (Biological Oxygen Demand) has been excluded from analysis.

6. PLOS authors have the option to publish the peer review history of their article (what does this mean? ). If published, this will include your full peer review and any attached files.

**Do you want your identity to be public for this peer review?** For information about this choice, including consent withdrawal, please see our Privacy Policy .

Reviewer #1: No

---

## [Author Response · Author response to Decision Letter 1]

22 Apr 2025

Dear reviewer,

Thank you very much for your kindly comments on our manuscript. There is no doubt that these comments are valuable and very helpful for revising and improving our manuscript. In what follows, we would like to answer the questions you mentioned and give detailed account of the changes made to the original manuscript. The new additions and additions to the revised parts, we have highlighted them in yellow in the revised original draft to present them more clearly.

Q1. Authors should define green agriculture in introduction section as the green agriculture is mentioned in the title.

Response: Thank you for your suggestion. Due to our negligence, we were unable to provide an overview of green agriculture in the summary. Our modifications are as follows. Firstly, in lines 10 -12 of the text, the following has been added: “Food security is the lifeline of national security. It is not only an important cornerstone for world peace, stability, and development, but also the core driving force for promoting the green development of agriculture." Besides, Lines 20 to 24 in the text once again elaborate that suppressing agricultural non-point source pollution can contribute to ensuring food security and promoting the green development of agriculture.

Q2. Sentence mentioned in the introduction (first page) In 2021, the net per-acre usage of nitrogen, phosphorus, and potassium compound fertilizers for China's three major grain crops was 8.39 kg, 0.44 kg, 0.56 kg, and 15.54 kg, respectively needs to be check again. it seems some typographical error.

Response: Thank you for your suggestions. Due to our negligence, the text in this paragraph is not expressed clearly, and there are errors in the unit printing. After consulting the corresponding literature and materials, we made changes to this paragraph. The specific changes are as follows: In lines 36 to 42 of the text, we further explain the usage amount of compound fertilizer containing nitrogen, phosphorus, and potassium in each mu of farmland, and the unit is kilograms instead of kilograms.

Q3. A total 11 tables have been presented in the paper, therefore authors should present one or two table in figure format.

Response: Thank you for your reminder. After the discussion, we have made the following modifications: Firstly, we have merged the previous Tables 6 and Table 7 into the current Table 6, which means that heterogeneous regression is placed in the same table for discussion. Furthermore, we merged Table 9 and Table 10 to form a new table 8, with the aim of placing the Moran indices of ANSP and FS in a unified table for comparison and evaluation. This can compress the table volume to a certain extent.

Q4. Authors should justify why Total K (potassium) and BOD (Biological Oxygen Demand) has been excluded from analysis.

Response: Thank you for your suggestion. Due to our oversight, we overlooked the expressions of the two variables, K (potassium) and BOD (Biological Oxygen Demand). After discussion, we made the necessary modifications. Specific modifications are as follows: In lines 252-259 of the article, we did not explain why K and BOD were included as constituent elements of ANSP. The main reason lies in that "the core of agricultural non-point source pollution lies in controlling agricultural eutrophication. toxic substances and physical damage. However, due to problems such as weak migration and poor detection adaptability of K and BOD.”

---

## [Editor Report · Decision Letter 1]

5 May 2025

Achieving green agricultural development: Analyzing the impact of agricultural non-point source pollution on food security and the regulation effect of environmental regulation

PONE-D-24-54342R1

Dear Dr. LU,

We’re pleased to inform you that your manuscript has been judged scientifically suitable for publication and will be formally accepted for publication once it meets all outstanding technical requirements.

Kind regards,

Murli Dhar Meena, Ph.D

Academic Editor

PLOS ONE

Additional Editor Comments (optional):

The manuscript written very well manner and author responded every comments made by reviewer (S) and justified with suitable references.
---

## [Editor Report · Acceptance letter]

PONE-D-24-54342R1

PLOS ONE

Dear Dr. LU,

I'm pleased to inform you that your manuscript has been deemed suitable for publication in PLOS ONE. Congratulations! Your manuscript is now being handed over to our production team.

Kind regards,

on behalf of

Dr. Murli Dhar Meena

Academic Editor

PLOS ONE